# Thiol Reductases in *Deinococcus* Bacteria and Roles in Stress Tolerance

**DOI:** 10.3390/antiox11030561

**Published:** 2022-03-16

**Authors:** Arjan de Groot, Laurence Blanchard, Nicolas Rouhier, Pascal Rey

**Affiliations:** 1Aix Marseille Univ, Molecular and Environmental Microbiology (MEM) Team, CEA, CNRS, BIAM, F-13108 Saint Paul-Lez-Durance, France; laurence.blanchard@cea.fr; 2University of Lorraine, Inrae, IAM, F-54000 Nancy, France; nicolas.rouhier@univ-lorraine.fr; 3Aix Marseille Univ, Plant Protective Proteins (PPV) Team, CEA, CNRS, BIAM, F-13108 Saint Paul-Lez-Durance, France

**Keywords:** *Deinococcus*, thiol, reductase, peroxidase, cysteine, oxidative stress, thioredoxin, bacillithiol, protein redox status

## Abstract

*Deinococcus* species possess remarkable tolerance to extreme environmental conditions that generate oxidative damage to macromolecules. Among enzymes fulfilling key functions in metabolism regulation and stress responses, thiol reductases (TRs) harbour catalytic cysteines modulating the redox status of Cys and Met in partner proteins. We present here a detailed description of *Deinococcus* TRs regarding gene occurrence, sequence features, and physiological functions that remain poorly characterised in this genus. Two NADPH-dependent thiol-based systems are present in *Deinococcus*. One involves thioredoxins, disulfide reductases providing electrons to protein partners involved notably in peroxide scavenging or in preserving protein redox status. The other is based on bacillithiol, a low-molecular-weight redox molecule, and bacilliredoxin, which together protect Cys residues against overoxidation. *Deinococcus* species possess various types of thiol peroxidases whose electron supply depends either on NADPH via thioredoxins or on NADH via lipoylated proteins. Recent data gained on deletion mutants confirmed the importance of TRs in *Deinococcus* tolerance to oxidative treatments, but additional investigations are needed to delineate the redox network in which they operate, and their precise physiological roles. The large palette of *Deinococcus* TR representatives very likely constitutes an asset for the maintenance of redox homeostasis in harsh stress conditions.

## 1. Introduction

Bacteria belonging to the genus *Deinococcus* are extremely tolerant to radiation, desiccation, and other conditions that generate oxidative damage to biomolecules, including DNA and proteins. The first described member of this genus, *Deinococcus radiodurans*, isolated as a contaminant from canned meat that was supposedly sterilised by a high dose of ionising radiation, has been studied most extensively [1,2]. In the last decades, more than 90 other *Deinococcus* species have been isolated worldwide from different places and environments (e.g., air, water, desert soils). Many genes and several mechanisms contribute to this tolerance [3]. *Deinococcus* bacteria are able to repair massive DNA damage generated by radiation or desiccation [4,5], indicating efficient limitation or repair of damage to other biomolecules, such as DNA repair proteins and other enzymes, to preserve their function under these conditions. Indeed, proteins in *Deinococcus* were found to be much better protected against oxidative damage than those in radiation-sensitive species, such as *Escherichia coli* [6,7,8,9]. Both nonenzymatic (e.g., carotenoids, a high intracellular Mn^2+^/Fe^2+^ ratio, and antioxidant complexes containing Mn^2+^, phosphate, and peptides) and enzymatic (e.g., superoxide dismutases, catalases, peroxidases, thioredoxins) antioxidant systems are present in *Deinococcus* [3].

Thiol-dependent antioxidant enzymes, namely, thiol reductases (TRs), are ubiquitous enzymes harbouring catalytic cysteines that modulate the redox status of Cys and Met in partner proteins. Living organisms possess various thiol-dependent antioxidant systems displaying specific features and electron sources [10]. Among them, the thioredoxin (Trx) and glutathione (GSH) systems are the most represented and well characterised [11,12,13]. The Trx pathway is composed of NADPH, NADPH-dependent Trx reductase (TrxR), and Trx, a disulfide reductase supplying electrons to many types of partners and targets, including peroxiredoxins (Prxs) and methionine sulfoxide reductases (Msrs), which are involved in the scavenging of peroxides and preservation of protein redox status, respectively. In the GSH system, electrons are provided by NADPH and transferred via NADPH-dependent GSH reductase to GSH, a low-molecular-weight (LMW) compound having direct antioxidant activity. GSH is also able to transfer reducing power to glutaredoxins (Grxs), proteins related to Trxs, and fulfilling a similar function in the control of Cys redox status in partner proteins. Trx and GSH systems play critical roles in the maintenance of protein redox homeostasis, particularly in responses to environmental constraints, upon diseases or during aging [10,14,15,16]. Indeed, these reducing systems participate in the prevention and repair of oxidative damage and in the control of redox post-translational modifications occurring in their partner proteins. This type of modification is associated with changes in enzyme activity, protein conformation or structure, and consequently with redox-based signal transduction pathways. While the Trx system is ubiquitous to all domains of life, the GSH system is absent in many bacteria, including *Bacillus* species and pathogenic bacteria, such as *Mycobacterium tuberculosis* or *Staphylococcus aureus*, in which thiol-based responses are crucial against host oxidative defences [17]. In these species, systems based on other LMW thiol compounds, such as mycothiol (MSH) or bacillithiol (BSH), have been identified and fulfil roles similar to those of GSH [18,19,20].

The extreme environmental conditions that *Deinococcus* is able to tolerate strongly affect among others cell redox homeostasis and thiol status. Thus, TRs are very likely part of the mechanisms underlying this outstanding feature [3]. Consistently, it was reported that a *Deinococcus geothermalis* mutant lacking the cystine importer system, which is a source of thiols, was more sensitive to an H_2_O_2_ treatment [21]. Some studies based on biochemical, pharmacological, or genetic approaches characterised the Trx system in *Deinococcus* [22,23,24] and indicated that it likely plays an essential role in responses to oxidative stress [25,26]. In a previous review describing *Deinococcus* antioxidant systems [3], the main TR families were listed, revealing that this bacterial genus possesses a complete Trx system, including a relatively high number of TRs related to Trxs and Prxs. Of note, *Deinococcus* does not possess GSH, but BSH and the relevant reducing system, including NADPH–BSH reductase (Bdr) and bacilliredoxin (Brx) [27].

In this review, we aim to carry out a more detailed and exhaustive description of TRs and related proteins among a representative set of various *Deinococcus* species, the genomes of which have been integrated into the MicroScope platform for microbial comparative genome analysis and functional annotation [28]. These species have been isolated from locations in North America, South America, Europe, Africa, and Asia (Table 1). The TRs and related proteins (locus tags) of these bacteria are listed in Table 2. The presence of most of these proteins has been reported in *D. radiodurans* and *Deinococcus deserti*, for which several shotgun proteomics studies have been performed [5,29,30,31,32,33]. We analyse the proteins notably with regard to their sequences, to the Cys distribution, and to the tridimensional structures and compare them with TRs from main model species (generally *E. coli* and/or *Bacillus subtilis*). We highlight the specificities of *Deinococcus* TRs and discuss based on available expression and phenotype data their physiological functions in line with the remarkable tolerance of this bacterial genus.

## 2. The TrxR/Trx System in *Deinococcus*

The two main components of the Trx system, TrxR, an NADPH-dependent flavoenzyme, and Trx, a small disulfide reductase, harbour conserved motifs, including two catalytic cysteines, allowing the transfer of reducing power via mixed-disulfide intermediates (Figure 1). This thiol-based system was initially discovered in *E. coli* through its role in DNA synthesis as an electron donor for ribonucleotide reductase [47]. Since then, this ubiquitous redox system, which regulates protein dithiol/disulfide balance using electrons provided by NADPH, was described as participating in a wide array of signalling and regulation pathways, and also in protective and repair mechanisms [10,15]. *E. coli* strains deleted for *trx* genes exhibit increased resistance to H_2_O_2_, probably because the more oxidising redox potential causes increased expression of catalase and alkyl hydroperoxidase in these mutants, but is more sensitive to diamide, a compound inducing disulfide bridging [48]. The key roles of the Trx system in the preservation of redox homeostasis and in responses to oxidative stress (Figure 1c) have been extensively documented in many prokaryotes and eukaryotes [10,11,15]. In *Deinococcus*, the Trx system has been more recently characterised, and in the next sections, we will describe the characteristics of TrxR and Trx families in this genus and what is currently known about their functions.

### 2.1. Deinococcus Thioredoxin Reductases

TrxR is a homodimeric flavoprotein belonging to the pyridine nucleotide–disulfide oxidoreductase family, which also includes enzymes such as glutathione reductase, alkyl hydroperoxide reductase, and lipoamide dehydrogenase [49]. Bacterial TrxRs, like those of archaea, fungi and plants, are composed of two subunits of 35 kDa, each possessing FAD-(flavin adenine dinucleotide) and NADPH-binding domains. These low-molecular-weight TrxRs are characterised by a specific fold found in several types of enzymes that catalyse diverse chemical reactions [50]. The electron transfer from NADPH via FAD to the active site disulfide, located in the NADPH-binding domain, leads to a large conformational change [51]. Bacterial TrxR-active sites contain two redox-active Cys within CATC or CAVC motifs in *E. coli* and *M. tuberculosis*, or *Bacillus anthracis* and *S. aureus*, respectively [10]. A search in the genomes of representative *Deinococcus* species revealed the presence of highly conserved TrxR enzymes, with more than 70% identity between them, and harbouring a CATC motif (Appendix A). Only one *trxR* gene is present in most *Deinococcus* species, as is the case in *E. coli* and *B. subtilis*. Remarkably, *D. geothermalis* and *D. peraridilitoris* have two *trxR* genes encoding very close proteins (98% and 91% identity, respectively). While *Deinococcus* TrxRs possess an active site sequence motif similar to that of *E. coli*, they share more overall homology with TrxR of *B. subtilis*. Thus, *D. radiodurans* TrxR (locus tag DR_1982) has 43% and 47% identity with *E. coli* and *B. subtilis* counterparts, respectively. *Deinococcus* and *B. subtilis* TrxRs do not possess extra Cys residues, such as the *E. coli* enzyme. Of note, *Deinococcus* TrxRs are characterised by the presence of a threonine just before the active site, this residue being an alanine or a tyrosine in TrxRs from *E. coli* or *B. subtilis*, respectively. There is some variability regarding the identity of this residue among bacterial TrxRs since a threonine is present in the *Helicobacter pylori* representative and a serine in the *M. tuberculosis* representative. Whether these residues modulate the activity of redox-active cysteines remains to be investigated.

In a pioneer work regarding the characterisation of the Trx system in *Deinococcus*, Obiero et al. [22] investigated the biochemical and structural features of TrxR of *D. radiodurans*. Using recombinant proteins, they showed that the reductase provides electrons to *D. radiodurans* Trx2 (DR_A0164) at the expense of NADPH, and also to *E. coli* Trx1, but with a lower affinity. The overall structure of *D. radiodurans* TrxR is very similar to that of TrxRs from other bacteria, with both NADPH- and FAD-binding domains containing variants of the canonical Rossmann nucleotide-binding fold. However, when compared with *E. coli* TrxR, some differences in the shape and charge distribution in the Trx-binding pockets were noticed and presumed to underlie the species-specific affinity previously mentioned [22]. This group further analysed the residues involved in TrxR–Trx interaction using computational alanine mutagenesis and by accurate analysis of the interface [23]. They concluded that four residues (M84, K137, F148, and F149) in *D. radiodurans* TrxR mainly account for the stability of the interface and the preferential binding affinity towards *D. radiodurans* Trx2. There is currently no genetic study unveiling the physiological function of TrxR in *Deinococcus*. In *D. radiodurans*, increased levels of the *trxR* transcript and TrxR protein have been detected after exposure to cadmium and UV/vacuum, respectively [32,52]. Using inhibitors of TrxR, Maqbool et al. [26] reported that *D. radiodurans* cells treated with auranofin or ebselen showed increased H_2_O_2_ sensitivity and increased levels of intracellular reactive oxygen species (ROS) and protein carbonyls compared with cells treated with H_2_O_2_ alone, indicating altered redox homeostasis when TrxR is inhibited.

### 2.2. Deinococcus Thioredoxins

TrxR supplies electrons to Trx, which is a small heat-stable disulfide reductase of ~10–12 kDa. Canonical Trxs carry a WCGPC-active site motif and share a conserved fold, the basic Trx-fold being composed of three α-helices surrounding four β-strands. The N-terminal catalytic Cys is present mainly as a thiolate, and thus can initiate a nucleophilic attack on oxidised Cys residues present in substrate proteins. This leads to the formation of mixed-disulfide intermediates and triggers deprotonation of the second Cys of the Trx-active site, named resolving Cys, which will catalyse the reduction of the heterodimer and the release of reduced substrate and oxidised Trx (Figure 1b). The *E. coli* genome, like many other bacterial species, contains two *trx* genes, *trxA* and *trxC*, which encode Trx1 and Trx2, respectively. In *E. coli*, a substantially higher level of expression of *trxA* has been reported compared with that of *trxC*. *E. coli* strains lacking both *trx* genes are still viable [48], very likely due to partial overlapping functions with the GSH-dependent system [13]. However, in other bacterial species, such as *B. subtilis*, *Rhodobacter sphaeroides*, and *Synechocystis* sp. PCC 6803, deletion of the *trxA* gene leads to loss of viability [53].

The characterisation of the *E. coli* Trx1 interactome in vivo has led to the identification of more than 250 proteins likely to be redox-regulated at the level of their Cys residues and participating in a wide diversity of cellular processes [54]. Similar conclusions, about the broad spectrum of Trx partners, have been drawn from the comprehensive characterisation of Trx interactome in other organisms, such as mammalian or photosynthetic cells [55,56]. Altogether, these findings clearly validate the central role of Trx in the control of most, if not all, cellular processes in living organisms and highlight its importance in redox biology [15].

The Trx1 isoform represents the most typical Trx form containing around 110 residues without any sequence extension or additional domain. Six of the analysed *Deinococcus* species possess a single *trxA* gene (Appendix A). Surprisingly, three *trxA* genes are present in *D. peraridilitoris*, two being located on a plasmid and encoding proteins more similar to each other (68% identity) than to the chromosome-encoded Trx1 protein. One of these plasmid genes, Deipe_3901, is located directly next to the second *trxR* gene (Deipe_3902) (Appendix A). *D. radiodurans* Trx1 (DR_0944) shares between 76% and 92% identity with Trx1 from other *Deinococcus* species (55% and 60% with the extra Trx1 from *D. peraridilitoris*) and 44% and 49% with Trx1 from *E. coli* and *B. subtilis*, respectively, revealing a closer relationship with the latter, as found for TrxR (Appendix A). Some specificity in *Deinococcus* Trx1 sequences can be noticed in the proximity of the active site motifs. Indeed, whereas the residues preceding the WCGPC-active site are well conserved in bacterial Trx1 proteins, the two subsequent ones differ in chromosome-encoded *Deinococcus* Trx1s (generally RI) compared with those (KM) observed in *E. coli* and *B. subtilis* Trx1 (Appendix A). This specificity might confer modified catalytic efficiency or altered access of the active site in *Deinococcus* proteins, but this needs to be further investigated by comparing the biochemical features of mutated recombinant proteins.

The *trxC* gene, encoding Trx2, is present only in three *Deinococcus* species (*D. radiodurans*, *D. deserti*, and *D. geothermalis*) analysed here. Trx2 is characterised by the presence of an additional N-terminal domain of 30 to 35 residues, including 2 other CXXC motifs [57] involved in zinc coordination [58]. This extension has been proposed to modulate Trx activity. Furthermore, if conditions of oxidative stress would affect zinc binding, the released zinc might contribute to a redox signalling pathway [59]. The three Trx2 isoforms from *Deinococcus* share between 51% and 66% identity and around 42% with *E. coli* Trx2 (Appendix A). Some differences are observed regarding the residue just preceding the WCGPC active site motif (T in *D. geothermalis* instead of P in other species) and in the one following this motif (V in *Deinococcus* Trx2 proteins instead of N in *E. coli* Trx2). Again, these specific features might be associated with changes in catalytic properties.

Similar to *E. coli*, a substantially higher expression of Trx1 compared with Trx2 has been observed in *D. deserti* under standard conditions, at both the mRNA and protein levels [31]. Trx1 is also more abundant than Trx2 in *D. radiodurans* [30,33], and both *trxA* and *trxC* showed enhanced expression following treatment with H_2_O_2_ [60]. Currently, there are only little data available regarding the biochemical properties and physiological functions of Trxs in *Deinococcus*. In their pioneer work on the Trx system in *D. radiodurans*, Obiero et al. [22] cloned the *trx* gene encoding the Trx2 protein (but improperly named Trx1 since it includes the sequence encoding the N-terminal extension). The recombinant protein catalyses reduction of insulin disulfide bridges, a typical feature of most Trx enzymes, and is reduced by *D. radiodurans* TrxR. Very recently, Kim et al. [60] provided data regarding the two Trx isoforms from *D. radiodurans*. First, they characterised their biochemical properties and showed that both display insulin and DTNB (5,5′-dithiobis(2-nitrobenzoic acid), a small-molecule disulfide) reduction activities in the presence of NADPH and TrxR. These activities were completely lost when active site Cys residues in Trx were replaced by Ser. Consistent with the regulatory role of the additional zinc-binding CXXC motifs in Trx2, substitution of Cys by Ser in these motifs impaired its reduction capacity. Of particular interest regarding Trx2, Kim et al. [60] determined its crystal structure and noticed that the orientation of the N-terminal Zn finger domain within the overall structure and with respect to the Trx-fold was clearly different in *D. radiodurans* from that observed in other bacterial Trx2 isoforms. They proposed that this unique feature could modulate protein–protein interactions, and be involved in the recognition of specific targets in this species. Finally, these authors also constructed *D. radiodurans* strains deleted for *trxA* or *trxC*, and showed that both mutants exhibit strongly reduced survival rates when treated with 60 mM H_2_O_2_. Interestingly, the mutant lacking Trx2 appeared to be more sensitive to this reactive oxygen species compared with that impaired for Trx1. Therefore, even though Trx2 is not common to radiation-tolerant *Deinococcus* species, it seems to contribute to stress tolerance in the species possessing this protein. These data, combined with those of Maqbool et al. [26] on TrxR, clearly show the key role of the TrxR/Trx system in the tolerance of *Deinococcus* to oxidative stress. No data about mutants lacking either *trxR* or both *trxA and trxC* have been reported, perhaps because such mutants are not viable.

### 2.3. Trx-Like Proteins in Deinococcus

The analysis of the *D. radiodurans* genome revealed the presence of several other genes (DR_2085, DR_A0072, DR_0057, DR_B0110, and DR_0948) encoding small proteins harbouring CXXC motifs, such a CPDC, CHLC, and CPGC (Table 3), and predicted to display a Trx-fold (Appendix A). These proteins are present in most *Deinococcus* species analysed here and divided in several subclasses (Table 2; Appendix A) [3,5]. Based on sequence homology, some of these proteins were first classified as glutaredoxin-like proteins. However, since Grxs generally display a CPYC active site motif and, most importantly, get their reducing power from glutathione, which is absent in *Deinococcus*, these reductases should rather be considered as Trx-like proteins. Consistently, DR_2085-type proteins contain a specific structural motif (Figure 2a), SGFRP, which is found in *E. coli* NrdH, a disulfide reductase possessing a canonical C(M/V)QC active site, participating in the reduction of NrdEF, a ribonucleotide reductase, and supplied with electrons by TrxR [61]. A deep analysis of the genomes of various *Deinococcus* species revealed variability in the number of genes encoding such Trx-like proteins: for instance, three in *D. deserti* and six in *D. gobiensis*. The DR_2085 and DR_0057 types, which have CPDC and CHLC active sites, respectively, are present and well conserved in all species selected (Figure 2a; Appendix A). For *D. radiodurans* and *D. deserti*, proteomics indicated low levels of expression of these proteins under standard growth conditions, but their detection might have been limited because of their small size. In the next parts, we will mainly focus on these two Trx-like types, notably in relation to the knowledge gained on related reductases in other organisms.

#### 2.3.1. Trx-Like Proteins Carrying a CPDC Active Site Motif

##### Thioredoxin-Related Proteins of 14 kDa

Among Trx-like proteins present in *Deinococcus*, the DR_2085 type is present in all species selected for this study and includes well-conserved members sharing between 74% and 92% identity with DR_2085 (Figure 2a). This Trx type is characterised by the presence of a WCPDC active site. Interestingly, this motif is typical of a specific family of Trx-like proteins named TRP14 for Trx-related protein of 14 kDa or Clot in plants [62]. TRP14, first characterised in mammalian cells [63], is present and well conserved throughout the animal kingdom as well as in plants [64,65,66]. TRP14-related proteins are also present in yeast, but are less represented in bacteria. One ortholog, encoded by locus HP_1458, was identified in *H. pylori* [67] and shown to be important for maintaining DNA and protein integrity, and for survival under oxidative stress conditions [68]. The WCPDC motif is also present in the thioredoxin-like proteins YdpP and YtpP from *B. subtilis*, and in YtpP (BA_4945) from *Bacillus anthracis* [69]. Compared with TRP14s from other organisms, *Deinococcus* TRP14s are shorter, since they do not display an N-terminal extension (Figure 2b). Furthermore, except for the WCPDC motif, sequence similarity is low between these proteins. Alignment of the sequences of the two TRP14-related proteins from *D. radiodurans* and *H. pylori* revealed 15% identity and 30% similarity, while 13% identity and 25% similarity exist between DR_2085 and human TRP14 (Figure 2b). Notably, in addition to the conserved WCPDC active site motif, these proteins share a (V/I)PTL(V/I/L) motif in their C-terminal part.

Mammalian TRP14 has been shown to get electrons from TrxR and displays very poor ability to reduce insulin and typical Trx substrates, such as ribonucleotide reductases, peroxiredoxins, or methionine sulfoxide reductases [63]. Consistently, *B. anthracis* TRP14 could reduce DTNB, but not insulin in the presence of NADPH and *B. anthracis* TrxR [69]. Note that *H. pylori* TRP14 exhibits disulfide reductase activity in both the DTNB and insulin assays [67]. Most interestingly, human TRP14 can efficiently reduce L-cystine, a source of cysteine [70], and also nitrosylated or persulfidated cysteines in proteins [70,71]. Based on these biochemical properties, TRP14 reductases are considered as main modulators in redox signalling pathways and could be involved in protection against metal stress or pathogenic viruses [66]. Whether the DR_2085-type proteins fulfil similar activities and physiological functions in stress responses of *Deinococcus* remains to be investigated. Induced expression of the DR_2085 gene and its Deide_06390 homolog has been observed after irradiation [31,72,73] and upregulation of DR_2085 upon cadmium exposure [52].

##### DR_A0072-Related Proteins

Another CPDC-containing Trx-like protein (DR_A0072) has been identified in *D. radiodurans*, with an Asp preceding this motif, that is present neither in Trx-like such as TRP14 nor in canonical Trxs. Homologs of DR_A0072 sharing between 61% and 70% identity are present only in three of the analysed *Deinococcus* species (*D. geothermalis*, *D. gobiensis*, and *D. proteolyticus*). Note that these homologs do not harbour a CPDC motif. Such a motif is present in another plasmid-encoded *D. proteolyticus* homolog (Deipr_2732), displaying less identity (41%) with DR_A0072 (Appendix A). Interestingly, the DR_A0072 gene is located between a gene encoding a heavy metal translocating P-type ATPase and a gene encoding an ArsR family regulator. This is similar in the three other *Deinococcus* species, suggesting a possible role of this Trx-like protein in the maintenance of metal homeostasis. Consistently, the DR_A0072 gene is upregulated after exposure of *D. radiodurans* to cadmium [52].

##### DR_B0110-Related Proteins

A CPDC motif is also present in DGo_PA0204 from *D. gobiensis*. However, this motif is not entirely conserved. Instead, a CPXC motif is rather present in homologs found only in three other *Deinococcus* species, including DR_B0110. The latter has 48% to 53% identity with Dgeo_1776, DGo_PA0204, and Deipr_2190 (Appendix A). Interestingly, in the four species, the DR_B0110 gene homologs are preceded by and probably in operon with *nrdIEF* gene homologs, suggesting that these Trx-like proteins could participate in the modulation of ribonucleotide reductase activity, thus being functional equivalents of NrdH proteins. Expression of the DR_B0110 gene was induced after exposure to gamma radiation [72,74].

#### 2.3.2. Trx-Like Proteins Carrying Other Active Site Motifs

##### DR_0057-Related Proteins

Another type of Trx-like proteins represented by DR_0057 in *D. radiodurans* is present in the seven *Deinococcus* species selected. DR_0057 shares between 41% and 65% identity with the six homologs. These proteins are 81 to 106 amino acids long (Appendix A) and possess a low similarity both with NrdH-redoxin family proteins and proteins possessing a glutaredoxin-like domain (DUF836). In the absence of the C-terminal SGFRP motif typical of NrdH-redoxins, they cannot be classified as such. Instead, they share a dicysteinic CxLC motif that resembles the one present in DUF836 members. With the notable exception of the *D. proteolyticus* representative (SCKLCQ motif), other members of the *Deinococcus* genus analysed exhibit a more conserved GCHLCE motif.

Little information is known among the few members of this class. The tridimensional structure of a related protein from *Mus musculus* has been deposited in the protein databank (pdb code 1WJK), but never published. The yeast protein was referred to as MGP12 (mitochondrial glutaredoxin-like protein of 12 kDa) because it was identified in a study focusing on the mitochondrial proteome of this organism [75]. However, this name obviously does not fit with the widespread presence in prokaryote organisms. Indeed, in addition to their presence in eukaryotes including fungi, plants, and metazoans, homologs are present in some bacterial and archaeal genera. The sequences are highly divergent, and for example, only 17% identity and 17% similarity exist between DR_0057 and *Arabidopsis* MGP12 (Figure 2c). In fact, only the cis-proline that is present in all Trx superfamily members and the CxLC motif are strictly conserved.

##### DR_0948-Related Proteins

Finally, another type of Trx-like protein is present in six out of the seven *Deinococcus* species analysed (no homolog in *D. peraridilitoris*), the corresponding protein in *D. radiodurans* being DR_0948. The sequence is well conserved, with DR_0948 sharing 64–86% identity with the proteins from the five other species (Appendix A). They also exhibit longer sequence (ca 150 residues) than that of canonical Trxs, but differ in the second residue of the active site motif (CSGC, CPGC, or CAGC). In *D. deserti*, the expression of the Deide_06780 gene is induced by radiation [31].

### 2.4. FrnE Oxidoreductase

FrnE subfamily proteins are predicted cytoplasmic thiol oxidoreductases belonging to the DsbA family (domain cd03024 in the Conserved Domain Database) [76]. They contain a CXXC motif in a DsbA-like thioredoxin domain. FrnE homologs are present in many bacteria, and many possess an additional conserved CXXXXC motif at the C-terminus. The name *frnE* was given for the first time to a gene present in a gene cluster (*frnA* to *frnU*) associated with the biosynthesis of the polyketide antibiotic frenolicin B in *Streptomyces roseofulvus* (GenBank accession AF058302.1). An *frnE* gene homolog is present in *Deinococcus* species but not located in an *frn* gene cluster as in *S. roseofulvus*. FrnE (DR_0659, drFrnE) from *D. radiodurans* has been studied. The expression of DR_0659 is upregulated after exposure of *D. radiodurans* to gamma radiation and desiccation [74], cadmium [52], and UV/vacuum [32]. Compared with the wild-type strain, a DR_0659 disruption mutant showed increased sensitivity in particular to cadmium and diamide, and less markedly to gamma rays and H_2_O_2_ [77]. Recombinant drFrnE showed disulfide isomerase and insulin reduction activities in vitro. In addition to the 22-CXXC-25 active site motif located near the N-terminus, drFrnE has a C-terminal tail harbouring a 239-CXXXXC-244 motif. Four different drFrnE crystal structures have been solved, with the crystals grown in the absence or presence of different reducing agents [78]. A disulfide bond between Cys22 and Cys25 was observed in the absence of a reducing agent. In all structures, the C-terminal tail of one molecule interacts with the active site of another drFrnE molecule, with the C-terminal CXXXXC located close to the active site CXXC. In crystals grown in the presence of DTT, an intermolecular disulfide bond was observed between Cys244 and Cys22. The C-terminal cysteines are not required for insulin reduction activity in the presence of DTT. The drFrnE protein could also reduce DTNB in the presence of NADPH and *E. coli* TrxR, but this activity was decreased when the C-terminal cysteines were mutated or deleted. Based on structural and biochemical data, the authors proposed that drFrnE is a cytoplasmic disulfide reductase and that the C-terminal CXXXXC is functionally important with a role in the regeneration of the active site [78]. DR_0659 shares 50–69% identity with FrnE from five species analysed here, but only 37% with FrnE from *D. maricopensis* (Appendix A). They all have the active site motif CP(W/F)C. Remarkably, FrnE from *D. maricopensis*, like *S. roseofulvus* FrnE, lacks the C-terminal CXXXXC motif (Appendix A), suggesting that the regeneration mechanism of the oxidised form is different for these homologs.

In addition to FrnE, two other predicted cytoplasmic proteins with DsbA-like domains are present in several *Deinococcus* species. One of these, belonging to the FrnE-like subfamily of the DsbA family (cd03025), is encoded by the genomes of *D. deserti* and *D. proteolyticus*. These two proteins (Deide_3p01230 and Deipr_1559) share 37% identity and have a CGWC or CIWC motif but no other Cys residues (Appendix A). Another protein with a DsbA-like thioredoxin domain, but with only a single conserved Cys residue, is found in five of the analysed species (e.g., DR_2335 in *D. radiodurans*) (Appendix A). To our knowledge, no experimental data concerning the function or expression have been reported for these proteins.

### 2.5. Periplasmic Disulfide Oxidoreductases

Dsb (disulfide bond) family proteins such as DsbA to DsbE and DsbG from *E. coli* are involved in the formation or reduction of disulfide bonds, and in the repair of oxidised Cys residues in extracytoplasmic proteins [14]. They have redox-active cysteines, including CXXC motifs. DsbA, DsbC, and DsbG are periplasmic proteins synthesised as precursors with an N-terminal signal peptide. DsbA catalyses the formation of disulfide bonds in proteins exported to the periplasm. DsbA is reoxidised by the cytoplasmic membrane protein DsbB. The DsbA and DsbB homologs in *B. subtilis* are called BdbD and BdbC, respectively, with BdbD attached to the membrane by an N-terminal transmembrane segment [79]. DsbC and DsbG are part of a periplasmic reducing system that repairs oxidised Cys of single-Cys-containing extracytoplasmic proteins [80]. DsbC also catalyses the isomerisation of incorrect disulfide bonds that are often introduced by DsbA, resulting in proper disulfide bonds and protein folding. DsbE, also called CcmG, reduces the disulfide bond that is formed in apocytochrome c in the periplasm to allow heme ligation. DsbE belongs to the TlpA-like family (domain cd02966 in the Conserved Domain Database) [76], also including TlpA (e.g., from *Bradyrhizobium japonicum*), ResA (*B. subtilis*), and similar proteins with a CXXC motif-containing Trx domain in the periplasm. DsbC, DsbG, and DsbE are maintained in the reduced state by the cytoplasmic membrane protein DsbD that catalyses the transfer of electrons from the cytoplasmic NADPH–TrxR–Trx system across the membrane. Many bacteria, such as *B. subtilis* and *H. pylori*, possess a shorter DsbD analog called CcdA [14,81].

*Deinococcus* species have several predicted Dsb proteins, but there is variation in the presence/absence and number of these proteins across the different species (Table 2). To our knowledge, these proteins have not been studied experimentally. The *B. subtilis bdbD* and *bdbC* genes are present in an operon (*bdbDC*). Similarly, two adjacent genes encoding DsbA and DsbB family proteins showing sequence similarity to *B. subtilis* BdbD and BdbC are present in six of the analysed *Deinococcus* genomes (Appendix A). Like *B. subtilis* BdbD, these DsbA family proteins (e.g., DR_0753) have a predicted N-terminal transmembrane segment and a KCPXCK active site motif (Appendix A). Unlike *B. subtilis* BdbD, but similar to *E. coli* DsbC, two additional conserved Cys residues are present, which might form a disulfide bond as in DsbC. *D. proteolyticus* has an additional more distant homolog (Deipr_2421) encoded by a gene on a plasmid and not adjacent to a *bdbC*-like gene. The BdbC (DsbB family) homologs (e.g., DR_0754) have a PCXLCW motif and two other conserved Cys residues (Appendix A). The same six *Deinococcus* species possess another (two in *D. radiodurans*: DR_2019 and DR_0560) DsbA-related protein having a predicted signal peptide, a QCPYC motif, and two other conserved Cys residues (Table 2; Appendix A). Experimental work is needed to establish whether these deinococcal DsbA family proteins have a role, like *E. coli* DsbA, in generating disulfide bonds in extracytoplasmic proteins or are functionally similar to DsbC or DsbG. Remarkably, DsbA and DsbB family proteins are absent in *D. peraridilitoris* (Table 2).

As reported previously [82], *bdbDC* homologs in several *Deinococcus* species are directly preceded, and possibly in operon [31], by a gene encoding a *Deinococcus*-specific membrane protein designated dCSP-1 and the gene encoding the DNA repair protein UvrA (Appendix A). It has been proposed that BdbD (DsbA) and, on the cytoplasmic side, UvrA may interact with dCSP-1, and that UvrA may receive electrons from BdbC (DsbB) to reduce oxidatively damaged Cys residues in CXXC motifs of UvrA, restoring its nonoxidised functional state [82]. However, as mentioned above, *D. peraridilitoris* does not have DsbA and DsbB proteins, but possesses UvrA and dCSP-1 (Appendix A).

Homologs of *E. coli* DsbD were not found in *Deinococcus*. However, at least one homolog of the DsbD analog CcdA is present in this bacterial genus. Except for the *ccdA* gene (DR_1300) in *D. radiodurans*, *ccdA* homologs are located in gene clusters encoding cytochrome c-type biogenesis proteins CcmA to CcmH in the other species, albeit with some variation in the genetic organisation (Appendix A). A *ccmG* (*dsbE*) gene is also present in these gene clusters. *D. deserti* and *D. peraridilitoris* have an additional two-gene cluster encoding homologs of CcdA and CcmG (DsbE) (Appendix A). The CcdA proteins have two conserved Cys surrounded by other conserved residues (Appendix A). The CcmG (DsbE) proteins contain a WXXWCXXC motif as found in TlpA-like family proteins from other bacteria, such as *E. coli* DsbE and *B. subtilis* ResA (Appendix A). The CXXC motif of the DsbE homolog present in each *Deinococcus* contains a Pro (CXPC) as in ResA and in the uncharacterised YneN of *B. subtilis*. The additional DsbE homologs from *D. deserti* (Deide_2p00420) and *D. peraridilitoris* (Deipe_0660) have a divergent CLVC motif. Deinococcal CcdA and DsbE (CcmG) proteins may be required only for cytochrome c maturation (reduction of disulfide bond in apocytochrome), but could also participate in the reduction of other substrates in the periplasm, and have a role in defence against oxidative stress, as reported for such proteins in other bacteria [83,84,85,86]. Alternatively, and in particular for additional proteins in *D. deserti* and *D. peraridilitoris*, DsbE might function as an oxidant instead of a reductant, as observed for DsbE from *M. tuberculosis* [87], and be involved in the formation of disulfide bonds since, as described above, no DsbA family proteins were found in *D. peraridilitoris.*

## 3. Main Trx Partners in *Deinococcus*

In addition to their roles in the modification of the structure or activity of numerous partners via redox post-translational modifications, Trxs supply electrons to several types of thiol reductases and peroxidases that have critical roles in the maintenance of redox homeostasis. Among them, two main types of enzymes, methionine sulfoxide reductases (Msrs) and peroxiredoxins (Prxs), control the redox status of proteins and scavenge peroxides, respectively (Figure 1c). Both enzyme types fulfil their biochemical function thanks to catalytic redox-active cysteines.

### 3.1. Methionine Sulfoxide Reductases

Oxidation of Met results in the formation of Met sulfoxide (MetO) and Met sulfone (MetO_2_), the latter modification being irreversible. MetO proportion can reach up to 40% of total Met during acute oxidative stress conditions [88]. MetO is mainly reduced back to Met by Msrs [89]. Oxidation of Met generates *S*- and *R*-MetO diastereoisomers, and two Msr types, A and B, specifically reduce these isomers, respectively. Although sharing a very similar thiol- and sulfenic acid-based redox chemistry, they do not exhibit any sequence similarity [90,91]. Of note, MsrA is able to reduce both peptide-bound and free MetO forms, in contrast to MsrB, which reduces the peptide-bound form in a more efficient way [92]. Most Msrs achieve their activity thanks to two redox-active Cys and a three-step catalytic mechanism following MetO reduction: (i) formation of a Cys sulfenic acid intermediate, (ii) formation of an intramolecular disulfide bond, and (iii) disulfide reduction generally by Trx and regeneration of activity [91,93]. Note that the regeneration of some Msrs having only one catalytic Cys is ensured by other thiol-containing compounds, such as Grxs [93,94].

Thiol-utilising Msrs are present in most living organisms, except in a few eubacteria and in one group of archaea [95]. They fulfil essential functions in stress tolerance and during ageing through modulation of the Met redox status in proteins [96,97]. In pathogenic bacteria, Msrs are considered essential determinants underlying their virulence and capacity to counteract host defences [98]. Thus, modification of the expression of *msr* genes leads to impaired responses to oxidative stress notably due to defective protein repair [99]. Consistently, Henry et al. [100] reported that the ubiquitous recombinase RecA, which promotes DNA recombination and induction of the SOS repair system, is a substrate of both Msr types. Indeed, oxidation of two Met residues in *E. coli* RecA is associated with dramatic changes in DNA recombination activity and SOS induction level, and the Msr system allows preserving the pool of functional RecA. Besides a repair function, these enzymes have also been proposed to play a direct antioxidant function by eliminating ROS via cyclic oxidation of Met in proteins [101]. Most importantly, a lot of evidence highlighted that the control of Met redox status is involved in the transduction of redox signals [102,103,104]. For instance, exposure to HOCl of *E. coli* cells results in activation via Met oxidation of the HypT transcription factor and enhanced expression of genes involved in protective mechanisms against this harmful agent [105].

In *Deinococcus*, the knowledge regarding the maintenance of Met redox status has only recently been explored. When submitting *E. coli* and *D. radiodurans* cells to ionising radiation (1000 Gy), Bruckbauer et al. [106] observed substantially greater oxidative modifications in *E. coli* proteome. Of note, methionine sulfoxide formation was the most frequent modification in *E. coli*, for which 137 instances were counted, while only one was detected in *D. radiodurans*. In another proteomic-based analysis, Chang et al. [107] exposed these two bacteria to 6700 Gy, a radiation level lethal for *E. coli* and causing damage in *D. radiodurans*. A much lower level of carbonylation was measured in the latter. These authors investigated whether protein-intrinsic properties could underlie this differential tolerance and noticed that in permissive conditions Met was less prevalent in *D. radiodurans* than in *E. coli*. In contrast, following irradiation, Met prevalence decreased in the latter, but strongly increased in the former, suggesting the occurrence of a more efficient Msr system under oxidative stress in *D. radiodurans* [107]. Consistently, using structure- and sequence-based models, *E. coli* MsrB was predicted to be intrinsically more sensitive to carbonylation than its *Deinococcus* counterpart. These data corroborate the likely importance of Msr proteins in the tolerance of *Deinococcus* species to severe stress conditions.

#### 3.1.1. MsrA Proteins in *Deinococcus*

Similar to most non-photosynthetic species, *Deinococcus* has only one *msrA* gene. MsrA proteins from the seven species analysed here share 58% to 77% identity and about 45% identity with those of *E. coli* and *B. subtilis* (Appendix A). They all display four conserved Cys, which are at positions 25, 28, 60, and 170 in DR_1849, the Cys25 in the conserved motif A(I/L/F)GGCFWCT being the catalytic one. This motif is close to the consensus sequence present in most MsrAs [91,108]. Of note, this motif displays some specificities in *Deinococcus*, which are not observed in *E. coli* and many other organisms. For instance, it includes a second Cys at position 28, whereas Gly or Ser is generally present at this position. Interestingly, this second Cys is also present in *B. subtilis* MsrA. In *E. coli* MsrA, Cys198 has been identified as the resolving Cys allowing activity regeneration of the enzyme [109]. Based on sequence alignment, we presume that the resolving Cys in *Deinococcus* reductases is the one corresponding to Cys170 in DR_1849. Upregulation of DR_1849 gene expression or protein abundance has been observed after exposure of *D. radiodurans* to radiation [32,74] or cadmium [52].

#### 3.1.2. MsrB Proteins in *Deinococcus*

Like in most prokaryote genomes [110], one *msrB* gene is present in *Deinococcus*. Among the seven species selected for this study, six MsrB proteins display high homology between them (72% to 82% identity) (Appendix A). MsrB (Deipe_4299) from *D. peraridilitoris* is more divergent since it exhibits only 51% to 55% identity with MsrBs from other species. *Deinococcus* MsrBs (except Deipe_4299) are closer to the one of *B. subtilis* (ca. 62% identity, against ca. 48% when compared with *E. coli* MsrB). Interestingly, *D. peraridilitoris* MsrB possesses two CXXC motifs like homologs from *E. coli* and eukaryotes [108,111]. These motifs are presumed to be involved in Zn fixation and structural stabilisation of the protein, and are absent in the reductases of the six other *Deinococcus* species analysed here and of *B. subtilis*. In other respects, all *Deinococcus* MsrBs share two well-conserved Cys, at positions 71 and 126 in DR_1378, corresponding to Cys63 and Cys117, respectively, in *E. coli* MsrB. Analysis of the MsrB catalytic mechanism [112] revealed that the latter, which is included in the well-conserved active site motif RYC(I/V)N, corresponds to the catalytic Cys, while the former named resolving Cys is present in the motif GCGWP. Based on these sequence features, we conclude that *Deinococcus* MsrBs belong to the 2-Cys type, which is regenerated by Trx following the reduction of an intramolecular disulfide bridge [113]. Induced expression of the DR_1378 gene was observed after exposure of *D. radiodurans* to cadmium [52], while its gene homolog Deide_04050 in *D. deserti* was induced after gamma radiation [31].

#### 3.1.3. Non-Thiol-Based Periplasmic and Possible Cytoplasmic Msr Systems

As described in Section 2.5, the maintenance of Cys redox status in periplasmic proteins in bacteria is ensured by the thiol-dependent Dsb system thanks to reducing power provided by cytoplasmic Trx. In contrast, non-thiol-dependent systems control the redox status of Met in periplasm [114], except in some bacteria, such as *Neisseria* or *Helicobacter*, in which Msrs A and B are present in this compartment as fusion proteins [115,116]. In most other bacteria, various types of molybdenum-containing enzymes exist to reduce MetO [114]. The MsrPQ (YedY/YedZ) system is present in a large subset of bacteria including *E. coli* and *Pseudomonas aeruginosa* [117]. In this system, electrons are provided by the respiratory chain via MsrQ, an inner membrane heme *b*-containing protein, to MsrP, which reduces MetO diastereoisomers thanks to a molybdopterin cofactor. Both MsrP and MsrQ are present and well conserved in all *Deinococcus* species analysed here (Table 2), the *D. radiodurans* representatives being DR_2536 and DR_2537, respectively. For DR_2536, upregulated gene expression has been observed following exposure to cadmium [52].

Other molybdenum-containing enzymes exhibiting MetO reductase activity are the cytoplasmic BisC and periplasmic TorZ and DmsA proteins present in *E. coli* and other bacteria. These are oxidoreductases containing the COG0243 or BisC domain [114]. Periplasmic homologs of this enzyme type are not present in *Deinococcus*. However, a predicted cytoplasmic protein (DR_0397 in *D. radiodurans*; Table 2) with only low levels of similarity (less than 30% identity) with BisC is conserved in *Deinococcus*. Furthermore, another predicted cytoplasmic molybdenum-containing enzyme belonging to the sulfite oxidase family and containing the COG2041 (YedY) domain also present in MsrP is highly conserved in *Deinococcus* (DR_0716 in *D. radiodurans*; Table 2). Whether these uncharacterised proteins have MetO reductase activity or another function remains to be established.

### 3.2. Thioredoxin-Dependent Peroxidases/Peroxiredoxins

Peroxiredoxins are ubiquitous enzymes reducing peroxides using a conserved cysteine, named peroxidatic cysteine [118,119]. These enzymes, which are generally highly abundant in cells, function without any cofactor and exhibit a remarkable efficiency in scavenging peroxides and peroxynitrites [118]. They are recognised as key actors in the maintenance of cell redox homeostasis via several functions, including elimination of peroxides, signalling due to hyperoxidation or via redox modulation of protein partners, and also a chaperone role in relation to their redox status and oligomerisation level [119,120,121]. Since their discovery in yeast [122], Prxs have been the subject of extensive work in all organisms. They are critical actors in responses to environmental constraints, upon diseases such as cancer, or during the course of ageing [118,119]. They also constitute essential virulence factors in pathogenic microorganisms [123].

Prxs are classified in six evolutionary distinct subfamilies (Prx1, Prx6, AhpE (alkyl hydroperoxide reductase E), PrxQ/BCP (bacterioferritin comigratory protein), Tpx (thiol peroxidase), and Prx5), which differ by the presence and position of resolving cysteine, the oligomeric state, and the type of electron donor [118]. All are characterised by the presence of a conserved motif, Pxxx(T/S)xxC, containing the peroxidatic Cys, the thiolate form of which attacks peroxides. The resulting Cys sulfenic acid is generally resolved by the formation of an inter- or intramolecular disulfide with a resolving Cys, which will be further reduced by Trx in most cases, except for AhpE and AhpC, a member of the Prx1 subclass. AhpC-type Prxs receive electrons from AhpF, a flavoprotein pyridine nucleotide–disulfide oxidoreductase related to, but different from, TrxR [124].

The survey of *Deinococcus* genomes revealed the presence of two types of Prxs, namely, PrxQ/BCP and AhpE (Figure 3), which are present in all analysed species of the genus [3] (Table 2). In this section, we will focus on the Trx-dependent PrxQ/BCP type, first identified and characterised in *E. coli*, and named bacterioferritin comigratory protein in bacteria [125], but PrxQ in plants [126]. In *E. coli*, the *bcp* gene is induced by oxidative stress conditions, and strains lacking it exhibit hypersensitivity to hydrogen and organic peroxides [125]. The first characterised bacterial or plant representatives of this class harboured two Cys, forming an intramolecular disulfide bond following the catalytic act, which was subsequently reduced by Trx [126,127]. However, sequence alignment of BCP proteins from prokaryotes revealed the absence of the resolving Cys in a majority of them and raised the question of the regeneration of their activity [127]. When investigating the catalytic mechanism of 1-Cys BCP from *Burkholderia cenocepacia*, Clarke et al. [128] observed that the formed Cys sulfenic acid could be reduced directly by Trx, or in a more efficient way by the GSH/Grx pathway.

With regard to *Deinococcus*, *D. deserti* and *D. proteolyticus* have only two *bcp* genes, whereas the other species possess at least three genes (Table 2). Among the three genes in *D. radiodurans*, DR_1208 possesses a single catalytic Cys, whereas the two other Prxs (DR_0846 and DR_1209) carry two redox-active Cys. DR_0846 is the closest homolog to *E. coli* and *B. subtilis* BCP proteins (40% and 43% identity, respectively) with redox-active cysteines separated by four residues. The percentage identity among *Deinococcus* DR_0846-related BCPs (Appendix A) ranges from 64% to 77%. The remaining *Deinococcus bcp* genes also code for 2-Cys peroxidases, but about 38 residues (Appendix A) separate the two conserved Cys. DR_1209 shares 45% to 74% identity with the DR_1209-related BCPs of the other species, and 48% with DR_1208. Of note, the Pxxx(T/S)xxC motif present in most BCPs is replaced by a Pxxx(G/A)xxC motif in 2 (out of 11) of these atypical BCPs. Furthermore, the presence of extra Cys was noticed in five of these BCPs (Appendix A). These sequence features might alter the peroxidase catalytic properties and underlie specific interactions or regeneration pathways with electron donors. An additional, longer and more distant, predicted BCP-type Prx is present in *D. gobiensis* (Appendix A). This DGo_CA0314 protein shares 64% identity with *D. deserti* Deide_23291, but the latter seems truncated. Both DGo_CA0314 and Deide_23291 may represent inactive BCPs since they lack the PxxxTxxC motif, and a 15-residue insertion is present in the corresponding region.

Cho et al. [25] investigated the expression levels of the three *bcp* genes in *D. radiodurans* exposed to oxidative stress generated by H_2_O_2_ or subjected to gamma irradiation. DR_0846 was strongly induced by both treatments, while the expression of DR_1209 was moderately increased by the peroxide, and that of DR_1208 was substantially triggered by irradiation. In proteomics studies, DR_0846 (and its *D. deserti* homolog Deide_10900) was well detected compared with DR_1209 and DR_1208 (and Deide_09051), suggesting higher expression levels of the DR_0846-type BCP under standard conditions [5,29,30,31,32,33].

The *D. radiodurans* DR_0846 BCP protein has been further characterised with regard to its activity and physiological function [25]. The recombinant protein exhibits peroxidase activity towards H_2_O_2_ and holdase chaperone activity when measuring thermal-induced aggregation of a heat-sensitive substrate. Interestingly, a *D. radiodurans* strain lacking DR_0846 displays similar sensitivity to H_2_O_2_ and heat stresses compared with WT. Most importantly, a strain lacking both DR_0846 and catalase exhibited increased sensitivity to these stresses compared with the strain lacking only catalase, indicating that the BCP peroxidase activity in *Deinococcus* may represent a second line of defence towards oxidative stress. In other respects, BCP proteins from various bacterial species have been reported to bind DNA in vitro and protect it against thermal and oxidative damage [129,130]. The physiological relevance of such a function remains to be established, but in the case of *Deinococcus*, where DNA damage is repaired in an efficient and rapid manner following extreme stress conditions, we can propose that BCPs, which are over-represented in this species, participate in such protective mechanisms.

## 4. Thioredoxin-Independent Thiol Peroxidases

Besides Trx-dependent Prxs, other less ubiquitous types of thiol peroxidases have been identified in the last decades. Among them, three are present in *Deinococcus*, namely, AhpE, AhpD, and OsmC/Ohr.

### 4.1. The AhpF/D-AhpC/E Thiol-Dependent Peroxidase Systems

The alkyl hydroperoxide reductase AhpF-AhpC system has been first identified in *Salmonella typhimurium*, and later in many other bacteria, yeast, and mammals [131,132,133]. This system, catalysing the reduction of peroxides at the expense of NADH, is composed of AhpF, a FAD-containing protein, which mediates electron transfer to the AhpC thiol peroxidase [124]. As already mentioned, AhpF, like TrxR, belongs to the family of pyridine nucleotide–disulfide oxidoreductases. Both share a domain harbouring two conserved catalytic Cys [131,134]. However, AhpF has a higher molecular mass (57 kDa) due to the presence of an N-terminal extension of 200 residues, including two other Cys ensuring electron transfer from the TrxR-like domain to AhpC [135]. AhpC is a 21-kDa thiol peroxidase belonging to the Prx1 subfamily initially named typical 2-Cys Prx [118]. These highly expressed Prxs function as dimers and form during the catalytic cycle an inter-subunit disulfide bond. Further, as a function of their redox status, they are able to oligomerise and fulfil then a molecular chaperone function [136,137]. The AhpF–AhpC system is widespread in microorganisms, and strains modified for its expression exhibit altered susceptibility to oxidative stress generated by hydrogen or organic peroxides [133,138].

AhpE is an atypical type of AhpC having only one Cys and possessing also both peroxidase and chaperone activities [139]. It has been first characterised in *M. tuberculosis*, where it shares 34% sequence identity with AhpC [140]. Most interestingly, AhpE has been found to display specificity towards long-chain fatty acid hydroperoxides [141]. Interestingly, another type of reductase named AhpD is able to provide electrons to AhpC [142]. AhpD shares little sequence homology but common structural features with AhpF [10]. This reductase is present in some bacteria (*M. tuberculosis*, *Corynebacterium glutamicum*) and cyanobacteria (*Anabaena* PCC7120) [143]. AhpD is reduced by a lipoamide-containing protein, dihydrolipoamide succinyltransferase or lipoamide reductase, using NADH as a source of electrons [142]. AhpD plays essential roles in response to oxidative stress [143,144] that could be carried out by direct peroxidase activity towards organic peroxides [144] or reduction of AhpC/AhpE [10,142]. However, the experimental evidence for the latter biochemical function remains to be firmly established regarding AhpE.

In *Deinococcus* genomes, there are no close homologs of *ahpF* and *ahpC*. However, all species possess one *ahpE* gene (DR_2242 in *D. radiodurans*). DR_2242 shares 53% to 82% identity with the homologs from the other six species and 35% identity with AhpE from *M. tuberculosis* (Appendix A). In addition to the AhpE homolog Deipe_1016, the *D. peraridilitoris* genome encodes Prx-like protein Deipe_4202, which has less than 30% identity with the deinococcal AhpE homologs. Unlike other AhpE proteins, Deipe_4202 has a CxxC motif and lacks the PxxxT/SxxC motif. In *M. tuberculosis*, which does not possess GSH but another LMW thiol compound referred to as mycothiol (MSH), the reduction of AhpE is ensured by a MSH- and/or mycoredoxin-dependent mechanism [145]. This raises the question of the reducing system for AhpE in *Deinococcus*, in which the bacillithiol system is present. In other respects, all *Deinococcus* species harbour one *ahpD-like* gene (DR_1765 in *D. radiodurans*), two and five being present in *D. deserti* and *D. peraridilitoris*, respectively. Four of the *ahpD* genes of *D. peraridilitoris* are located on a plasmid, three being adjacent or close to extra *trxR* and *trx* genes in this species (Appendix A). However, these proteins show substantial variation in their sequences. DR_1765 shares 72–76% identity with Deide_13030, Dgeo_1446, and DGo_CA1027; 54% with Deima_0298; 36% with Deipe_3296; but only 17–25% with the other deinococcal AhpD-like proteins (including the single AhpD-like from *D. proteolyticus*) and 12% with AhpD from *M. tuberculosis* (Appendix A).

Biochemical and structural investigations of the *D. radiodurans* AhpD-like protein were performed by Zhao et al. [146], who reported that a mutant strain lacking the reductase exhibits increased sensitivity to H_2_O_2_. Of note, complementation assays using wild-type AhpD revealed partial restoration of this phenotype, and this level of restoration was much lower when using AhpD variants mutated for the two Cys of the conserved CxxC motif [146]. In other respects, determination of a crystal structure revealed that it adopts a fold representative of the AhpD superfamily [146]. Taken collectively, these data strongly support a role of AhpD and AhpE enzymes in *Deinococcus* responses to oxidative stress via peroxidase activity for both or possibly through electron supply from AhpD to AhpE.

### 4.2. Ohr/OsmC/YhfA Thiol-Dependent Peroxidases

Another class of bacterial thiol peroxidases named Ohr (organic hydroperoxide resistance) has been identified by complementation with *Xanthomonas campestris* genes of *E. coli* strains deficient in the AhpF–AhpC system and susceptible to organic hydroperoxides [147]. This 14-kDa protein shares some sequence homology (around 20% identity) with other stress-induced proteins termed OsmC (Osmotically induced protein C) first identified in *E. coli* [148]. Ohr and OsmC are structurally related enzymes [149]. Atichartpongkul et al. [150] reported that these two subfamilies are widely distributed in Gram-negative and Gram-positive bacterial species. Later, members of the Ohr/OsmC family have been identified in several eukaryotic clades, especially in fungi but also in mosses, such as *Physcomitrella patens* [151].

Ohr/OsmC proteins display specific expression patterns depending on stress nature and on peroxide type. In *P. aeruginosa*, *ohr* expression is induced only by organic peroxides, and not by other oxidative or stress treatments, while *osmC* expression is specifically triggered by salt treatment [150]. The phenotype of strains lacking these genes and the presence of two conserved Cys (one positioned in a typical VCPY motif in Ohr) in both protein types indicated that they could fulfil a thiol peroxidase function [150]. This role in metabolising hydrogen and organic peroxides was demonstrated in vitro using recombinant Ohr or OsmC forms and in vivo by generating deletion strains in several bacterial species [152,153,154,155]. Moreover, Ohr was shown to participate in response to peroxynitrite in *P. aeruginosa* [152]. Site-directed mutagenesis revealed that the two Cys are critical for the catalytic activity [154], and provided clear evidence that Ohr functions thanks to reactive thiol groups. Of note, these enzymes do not share structural relationship with Prxs, such as BCP, AhpC, or AhpE, as shown by the determination of crystal structures of *P. aeruginosa* Ohr and *E. coli* OsmC [149,154] (Appendix A). Following peroxide reduction, these enzymes form an intramolecular disulfide bridge that Trx- and GSH-based systems are unable to reduce, but require lipoylated proteins as an electron source (Figure 3) [156]. Indeed, *Xylella fastidiosa* Ohr was found to interact with lipoylated proteins (dihydrolipoyl dehydrogenase LpdA, pyruvate dehydrogenase beta PDHB, and dihydrolipoamide succinyltransferase SucB), and its peroxidase activity was supported by PDHB and SucB in the presence of lipoamide or by LpdA alone [156]. Similarly, OsmC is reduced by lipoyl-dependent systems [156]. In *C. glutamicum*, a bacterial species highly resistant to organic hydroperoxides, OsmC activity was preferentially regenerated through a pathway involving the AhpD reductase, another lipoyl-dependent system [157].

Another family of proteins structurally related to Ohr/OsmC has been described by Shin et al. [149]. It is present in *E. coli* (YhfA) but also in other bacterial species, such as *Vibrio cholerae*. YhfA proteins possess the two cysteines that are well conserved in Ohr and OsmC peroxidases and likely to participate in the catalytic mechanism, but they lack two residues, one Arg and one Glu, neighbouring the peroxidatic cysteine [149]. The biochemical properties and physiological roles of this class of putative thiol peroxidases remain to be investigated.

Among the *Deinococcus* species analysed in this review, most possess genes encoding these three types of atypical thiol peroxidases (Ohr, OsmC, and YhfA corresponding to DR_1857, DR_1538, and DR_1177 in *D. radiodurans*), except *D. deserti* and *D. proteolyticus*, which do not have *ohr* and *osmC* genes, respectively (Table 2). Note also that the genomes of several species contain two *ohr* or *yhfA* genes. *Deinococcus* OsmC proteins share 54–83% identity, and around 42% with the *E. coli* homolog (Appendix A). Concerning Ohr, *D. radiodurans* shares 55–78% identity with the closest homologs in the five other *Deinococcus* species, around 40% with the extra Ohr present in three of these species, and around 47% with the two Ohr proteins from *B. subtilis* (Appendix A). YfhA homologs are present in the seven *Deinococcus* species (Appendix A). They are divided into two subgroups, one containing highly conserved proteins in the seven species sharing 70–92% identity and around 35% with *E. coli* YhfA, and the second containing the two extra proteins from *D. deserti* and *D. peraridilitoris* having 71% identity between them but low-sequence identity with the other YhfA proteins (e.g., 21% between the two from *D. deserti*).

The knowledge about these atypical thiol peroxidases in *Deinococcus* remains limited. Atichartpongkul et al. [150] reported that in *D. radiodurans*, *osmC* (DR_1538) is induced by ethanol treatment and not by salt, while *ohr* (DR_1857) expression is enhanced by organic hydroperoxides. Cadmium exposure enhanced the expression of *osmC*, *ohr*, and *yhfA* in *D. radiodurans* [52]. Radiation-induced expression of *D. deserti osmC* [31] and *D. radiodurans* Ohr [158] has also been observed. The crystal structure of *D. radiodurans* Ohr has been solved [159] (Appendix A). Comparison with the structure of *P. aeruginosa* Ohr showed that Ohr proteins adopt two different conformations depending on the enzyme redox state. It was postulated that one configuration is consistent with efficient catalysis of the reduction of organic hydroperoxides, whereas the other form is required for enzyme recycling. Compared with wild-type *D. radiodurans*, a mutant strain deleted for *osmC* is more sensitive to gamma irradiation, particularly at high doses (10 kGy) [160], indicating the participation of OsmC peroxidase in the radiotolerance of *Deinococcus*.

## 5. The Bacillithiol System

Low-molecular-weight thiols are crucial for maintaining an intracellular reducing environment and for protection against ROS and other reactive species. Glutathione is the best-studied LMW thiol present, for example, in *E. coli* and eukaryotes. *Deinococcus* bacteria do not produce GSH but BSH, which is the most widespread LMW thiol in bacteria [161]. BSH is also involved in metal homeostasis [18]. The structure of BSH, the α-anomeric glycoside of L-cysteinyl-D-glucosamine with L-malic acid (Figure 4a), was determined after purification from *D. radiodurans* [162]. However, the BSH system has been studied most extensively in *B. subtilis* and *S. aureus* [18]. This system includes NADPH–BSH disulfide reductase (Bdr, previously called YpdA) and three bacilliredoxin (Brx) proteins (Figure 4b) [163,164,165]. Under oxidising conditions, BSH can form a disulfide (BSSB) and also mixed disulfides with protein thiols, a modification referred to as *S*-bacillithiolation. Like for *S*-glutathionylation and *S*-mycothiolation with GSH and MSH, *S*-bacillithiolation protects protein thiols from overoxidation, and may also function in redox regulation [19]. Brx proteins catalyse protein debacillithiolation, resulting in the formation of Brx-SSB. Oxidised Brx is likely preferentially reactivated by BSH or possibly by another Brx (see below) [18,165]. The regeneration of BSH from BSSB is catalysed by the NADPH-dependent Bdr enzyme [163,166] (Figure 4b). Recently, *B. subtilis* Bdr was shown to additionally have Brx reductase activity [165].

More than 50 proteins have been identified as *S*-bacillithiolated following hypochlorite or cumene hydroperoxide stresses in different *Bacillus* and *Staphylococcus* species. This includes glyceraldehyde-3-phosphate dehydrogenase (Gap), methionine synthase (MetE) and other amino acid biosynthesis enzymes, organic hydroperoxide resistance transcriptional repressor (OhrR), a predicted 2-Cys Prx (YkuU), translation factors, and chaperones [18,167,168]. The three Brx proteins identified in *B. subtilis* and *S. aureus* are named BrxA (previously YphP), BrxB (YqiW), and BrxC (YtxJ). BrxA and BrxB are paralogs and possess a redox-active CGC motif, whereas BrxC has only one Cys in a TCPIS motif. Debacillithiolation of Gap, MetE, and OhrR by BrxA and/or BrxB has been demonstrated [168,169]. Recently, BrxC was shown to debacillithiolate Gap, BrxB and Bdr, while BrxB was not debacillithiolated by BSH excess [165]. Similar to TrxR, Bdr is a FAD-containing NADPH-dependent oxidoreductase. While *Bacillus cereus* Bdr has a single-Cys residue, the Bdr proteins from *B. subtilis* and *S. aureus* have three-Cys residues, including one at position 14 shared with *B. cereus* Bdr (Appendix A). The characterisation of a *S. aureus* Bdr variant, in which Cys14 was replaced by Ala, suggested that Cys14 is the redox-active Cys as it seems required for BSSB reductase activity [163]. BSSB reductase activity was also demonstrated for *B. cereus* Bdr [164]. For *B. subtilis* Bdr, which could debacillithiolate BrxB-SSB and less efficiently BrxC–SSB, mutation of any of the three Cys residues led to decreased capacity to reduce both mixed disulfides [165]. A variant mutated for all Cys was essentially inactive. Of note, analysis of the crystal structures of Bdr from *S. aureus* and *B. cereus* indicated that Cys14 does not directly participate in the BSSB reductase reaction mechanism [164], and the same may be true for the Brx reductase activity [165]. In summary, the precise roles of these Cys in the BSSB or Brx reductase activities of Bdr remain to be delineated.

BSH is expected to be common to all members of the genus *Deinococcus* because the BSH biosynthesis genes *bshA*, *bshB*, and *bshC* are present in all species [3,27]. However, the other components of the BSH system, in particular Brx, appear to differ from their counterparts in *Bacillus* and *Staphylococcus*. Only a BrxC-related protein is present in *Deinococcus* species. This protein, containing a single Cys, is highly conserved (e.g., *D. radiodurans* DR_1832 has 71% to 82% identity with BrxC from the other analysed species), but displays low levels of similarity with BrxC from *Bacillus* and *Staphylococcus* (Appendix A). Instead of the TCPIS motif found in the latter, the corresponding sequence in deinococcal proteins is TCHKT. Moreover, *Deinococcus* BrxC proteins are substantially longer (204 to 219 residues instead of 108 residues for *B. subtilis* BrxC). The Trx-like domain in *Deinococcus* BrxC is located in the N-terminal part of the protein. The C-terminal region has some similarity with the bacterial self-protective colicin-like immunity domain (pfam09204). The function of this domain in *Deinococcus* BrxC is unknown. Because of the differences with BrxC from *Bacillus* and *Staphylococcus*, DR_1832 has been named AbxC (atypical BrxC) [27]. In *D. radiodurans*, expression of the *abxC* gene (DR_1832) is induced following exposure to cadmium [52]. A Bdr homolog is also present in all *Deinococcus*. *D. radiodurans* Bdr (DR_2623) has 74% to 89% identity with the homologs of the other six species and 40% with *B. subtilis* Bdr. DR_2623 contains three Cys residues (Cys22, Cys36, Cys275) that are conserved in deinococcal Bdr proteins, except for Deipe_2475, which lacks the first Cys corresponding to Cys22 of DR_2623 (Appendix A). Whether and how the Cys residues, located at different positions compared with those in Bdrs from *Bacillus* and *Staphylococcus*, support the catalytic mechanism of deinococcal Bdr is currently unknown. Upregulated expression of DR_2623 (Bdr) has been observed after exposure of *D. radiodurans* to cadmium [52], gamma radiation [158,170], and desiccation [171].

Experimental data obtained with *D. radiodurans* indicated that AbxC (BrxC) contributes to oxidative stress resistance [27]. In vitro, incubation of AbxC with H_2_O_2_ resulted in the formation of AbxC dimers through an intermolecular disulfide bond. NADPH consumption was observed when oxidised AbxC was incubated with BSH, NADPH, and Bdr (DR_2623), indicating reduction of the AbxC dimer by BSH and reduction of the formed BSSB by Bdr [27]. *S*-bacillithiolated proteins in *Deinococcus* have not been identified yet, and it is currently unknown whether the atypical AbxC can debacillithiolate proteins.

## 6. Conclusions

The remarkable tolerance of *Deinococcus* bacteria to extreme environmental conditions relies on a wide array of mechanisms ensuring efficient repair of DNA and limiting damage in proteins, many being present in all species. However, some genes, such as those coding for specific catalase types that confer resistance to oxidative stress, are not present in all species of the genus, revealing diversity in these mechanisms [3]. Regarding thiol reductases, most representatives are found in all *Deinococcus* species analysed here (Table 2), except Trx2, indicating that they are likely essential actors in the stress responses of this genus, as it is now well established in many other organisms. Consistently, deletion mutants for *trx*, *frnE*, *bcp*, *osmC*, *ahpD-like*, and *abxC* genes show increased susceptibility to oxidative treatments [25,27,60,77,146,160]. Nonetheless, the knowledge about the functions of *Deinococcus* thiol reductases remains limited. Thorough investigations need to be performed to delineate the precise physiological roles of these enzymes, notably by characterising mutants impaired in their expression in conditions of extreme stress, and by identifying their partners and substrates.

In addition to the NADPH-dependent Trx-and BSH-systems, *Deinococcus* possesses various types of thiol peroxidases using NADH as an electron source. Of note, Yamashiro et al. [172] reported a much greater concentration of NADH compared with that of NADPH in this genus, and a much higher NADH/NADPH ratio compared with *E. coli*, indicating that NADH is likely an essential actor in metabolism and stress responses in *Deinococcus*. Most interestingly, this genus displays a large diversity in thiol peroxidase types. On one hand, Trx-dependent BCPs are well represented in all *Deinococcus* species, since each possesses two or three genes coding for these Prxs, instead of one in *E. coli*. In addition, all *Deinococcus* species have two distinct types of NADH-dependent thiol peroxidases (i.e., AhpE/AhpD and Ohr/OsmC). The gene copy number and diversity of thiol peroxidases in bacteria, notably regarding the specialisation of their active sites, are likely associated with their level of stress tolerance [118,157]. Thus, AhpC, Ohr, and OsmC are widely distributed among bacterial species, but only some species simultaneously possess all of them, the others having generally one or two genes. For instance, Ohr is absent in *M. tuberculosis* and *E. coli*. The mycobacterium *C. glutamicum*, which is highly resistant to oxidative stress, possesses AhpD, Ohr, and OsmC homologs [143,157]. Regarding BCPs, Limauro et al. [173] identified three genes in the hyperthermophilic archaeon *Sulfolobus solfataricus* and proposed that this copy number was associated with the tolerance of this archaea to extreme temperature. The *Deinococcus* genus appears thus to be remarkably well equipped with a diversity of thiol peroxidases, since most species possess one or several *bcp*, *ahpD*, *ahpE*, *osmC*, *ohr*, and *yhfA* genes. This palette is very likely an asset in the extreme resistance of *Deinococcus* via efficient scavenging of all types of peroxides and peroxynitrites. Indeed, these thiol peroxidases display in vitro different specificity towards substrates [144,154,155,174]. For instance, OsmC is more efficient in metabolising organic peroxides than H_2_O_2_ [153]. In vivo investigations in *P. aeruginosa* showed increased susceptibility to artificial and fatty acid hydroperoxides of strains lacking Ohr, this phenotype being not restored by the expression of other types of thiol peroxidases, such as AhpC [152]. Taken collectively, these data strongly suggest that the diversity of thiol peroxidases in *Deinococcus* is associated with specialisation and efficiency regarding the scavenging of the various peroxide types produced upon stress conditions, and thus contributes to the outstanding tolerance of this genus.

The various electron transfer pathways involving thiol reductases present in *Deinococcus* are shown in Figure 5. Some pathways need to be precisely delineated, notably regarding the substrates of Trx and Brx proteins, which remain to be identified using biochemical and proteomics approaches. Based on the knowledge gained in other organisms, connections and interplays between these pathways very likely occur in *Deinococcus*. Indeed, in *E. coli*, there is functional redundancy between Trx and GSH systems [175], and an interplay between the two systems has been unveiled in plants during development [176]. Consistently, by generating *B. subtilis* strains impaired in BSH biosynthesis, Gaballa et al. [177] showed that this LMW thiol was not required to maintain the reduced status of protein thiols in the absence of stress, but noticed altered stress sensitivity of these strains. Regarding *Deinococcus*, it was recently reported that *D. deserti* and *D. radiodurans* mutant strains deficient in BSH biosynthesis are viable [27,59], indicating that in the absence of stress conditions the BSH system is not essential and likely compensated by the Trx system. Increased sensitivity to H_2_O_2_ has been reported for the *D. radiodurans* BSH-deficient mutant [27]. The occurrence of such compensation mechanisms and interplays is corroborated by the presence of Trx substrates or of other thiol reductases in bacillithiolated or Brx-interacting proteins. Among the bacillithiolated proteins identified in *Bacillus* and *Staphylococcus*, glyceraldehyde-3-phosphate dehydrogenase, a well-known Trx substrate [18,54,165,168], and a predicted 2-Cys Prx (YkuU) [167] have been found. The BSH system in *Deinococcus* is likely to allow the maintenance of the redox status and/or the regeneration of the activity of Trx-dependent enzymes, as shown for GSH in other bacteria [13]. In *Deinococcus*, Bihani et al. [78] proposed that in conditions of cadmium stress, which inhibits TrxR, Trx substrates could be supplied with electrons thanks to BSH. In *D. radiodurans*, 25 proteins potentially interacting with AbxC were isolated after incubation of cell extracts with AbxC-immobilised beads [27]. Among them, Ohr was identified, revealing interplays between the BSH/Brx system and NADH-dependent thiol peroxidases. BSH might also participate in the reduction of AhpE by analogy with the role of mycothiol. Indeed, in *M. tuberculosis*, reduction of AhpE can be performed via mycoredoxin- or mycothiol-dependent mechanisms [145]. Concerning lipoyl-dependent thiol peroxidases, a link has been unveiled in *C. glutamicum* between the AhpD system and Ohr/OsmC peroxidases, which in this species can be reduced by AhpD [157]. Taking into consideration all these data, we can thus hypothesise that the various pathways for electron transfer via thiol reductases and peroxidases in *Deinococcus* give rise to multiple connections and possible redundancy among them. This very likely allows the establishment of protective and repair mechanisms that can substitute or be switched on differentially depending on stress conditions, leading to the implementation of a complex and robust network (Figure 5) maintaining protein redox homeostasis in the extreme conditions that this bacterial genus can tolerate.

## Figures and Tables

**Figure 1 antioxidants-11-00561-f001:**
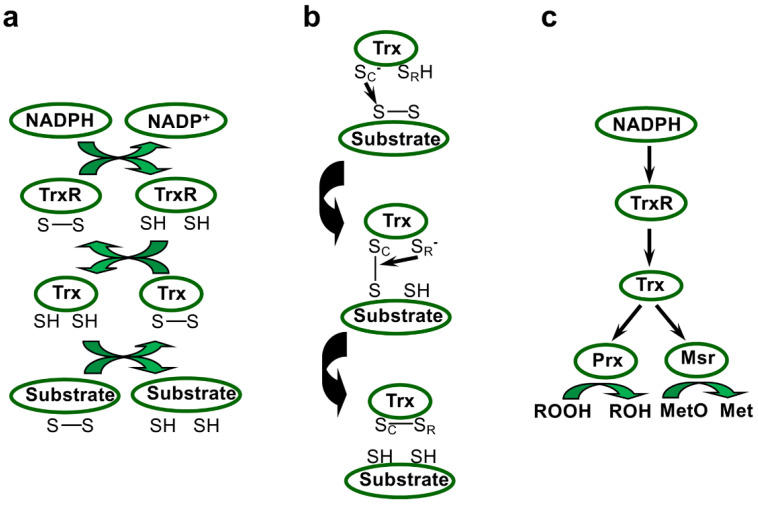
The Trx system. (**a**) NADPH-dependent electron transfer via Trx. (**b**) Catalytic mechanism of substrate reduction by Trx. (**c**) Trx thiol reductase substrates involved in the maintenance of cell redox homeostasis. TrxR, thioredoxin reductase; Trx, thioredoxin; Prx, peroxiredoxin; Msr, methionine sulfoxide reductase; S_C_H, catalytic cysteine thiol; S_R_H, resolving cysteine thiol; MetO, methionine sulfoxide.

**Figure 2 antioxidants-11-00561-f002:**
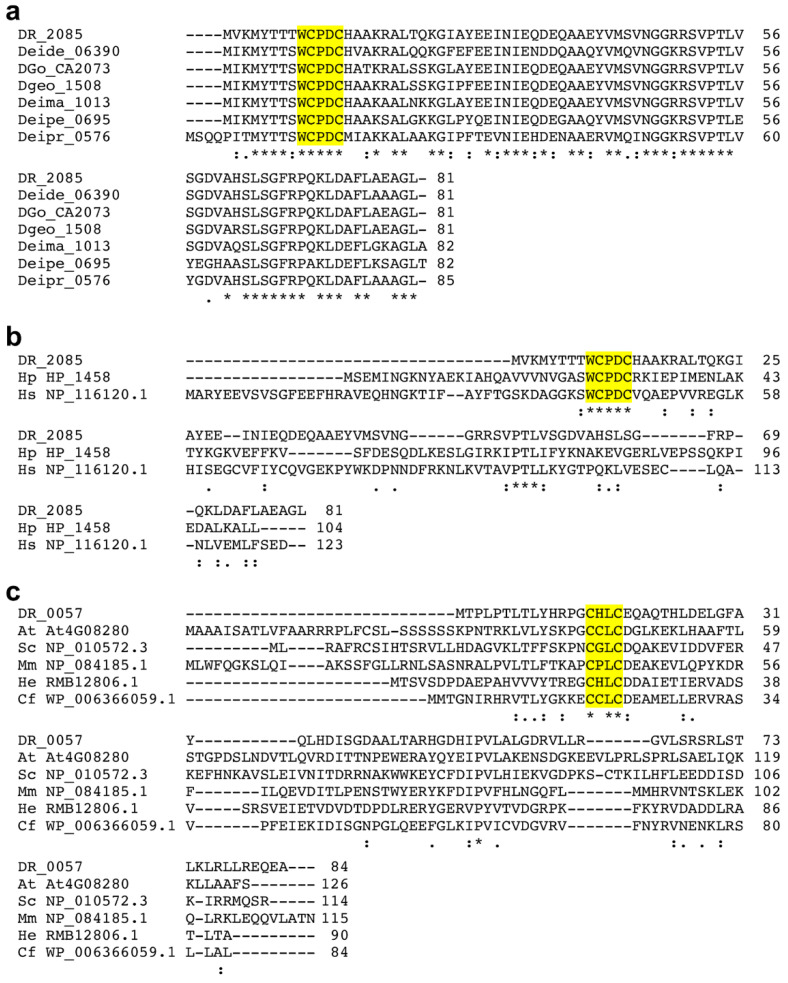
Multiple sequence alignments of TRX-like proteins from *Deinococcus*. (**a**) Alignment of *Deinococcus* DR_2085-type sequences. (**b**) Alignment of *Deinococcus* DR_2085 sequence with *Homo sapiens* (Hs) and *Helicobacter pylori* (Hp) homologous TRP14 Trxs. (**c**) Alignment of *Deinococcus* DR_0057 with *Arabidopsis thaliana* (At), *Mus musculus* (Mm), *Saccharomyces cerevisiae* (Sc), *Haloplanus aerogenes* (He) and *Chlorobium ferrooxidans* (Cf) homologous proteins. Trx active site sequences are highlighted in yellow. An asterisk (*) indicates positions which have a single, fully conserved residue; colon (:) and period (.) indicate conservation between groups of strongly and weakly similar properties, respectively. Abbreviations of *Deinococcus* species names are as specified in Table 1. Alignments were generated using UniProt ClustalO (https://www.uniprot.org/align/; accessed on 14 February 2022).

**Figure 3 antioxidants-11-00561-f003:**
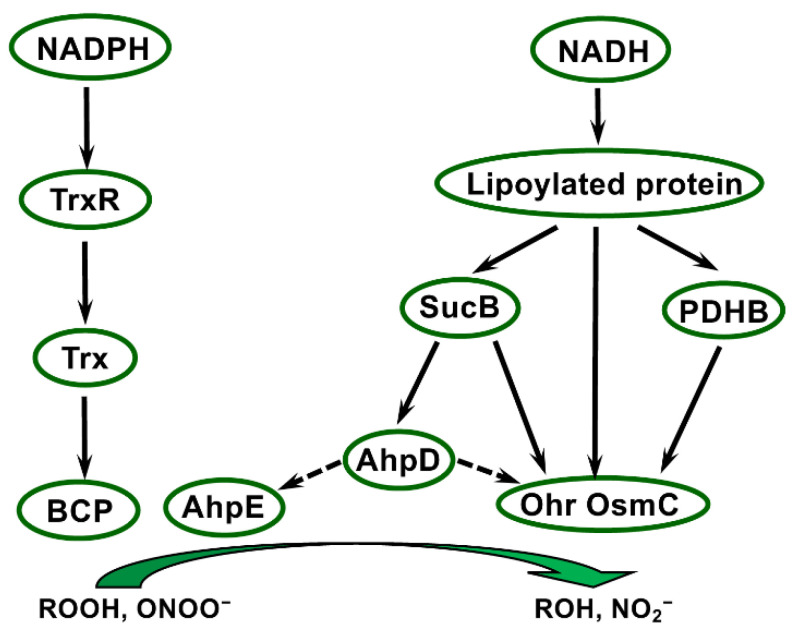
Electron sources for *Deinococcus* NADPH- and NADH-dependent thiol peroxidases (BCP, AhpE, AhpD, Ohr, OsmC) scavenging hydrogen and organic peroxides and peroxynitrites and participating in redox signalling. TrxR, thioredoxin reductase; Trx, thioredoxin; BCP, bacterioferritin comigratory protein; SucB, dihydrolipoamide succinyltransferase; PDHB, pyruvate dehydrogenase beta; Ahp, alkyl hydroperoxide reductase; Ohr, organic hydroperoxide resistance enzyme; OsmC, osmotically induced bacterial protein C. Dashed arrows indicate possible transfer of electrons.

**Figure 4 antioxidants-11-00561-f004:**
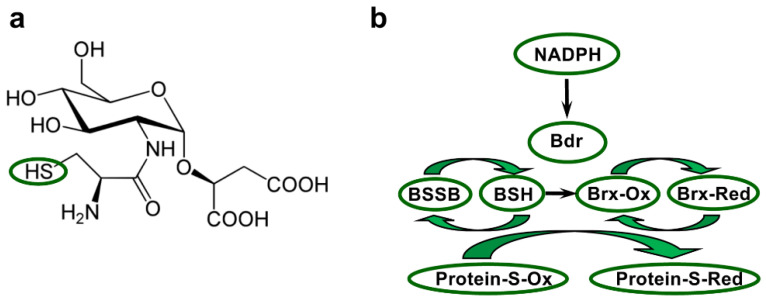
The bacillithiol system. (**a**) Structure of bacillithiol (BSH), glycoside formed of L-cysteinyl-D-glucosamine and malic acid. (**b**) Electron transfer pathways in the BSH system. Bdr, NADPH–bacillithiol disulfide reductase. Brx, bacilliredoxin; S-Ox, thiol oxidised (bacillithiolated) form; S-Red, thiol reduced form.

**Figure 5 antioxidants-11-00561-f005:**
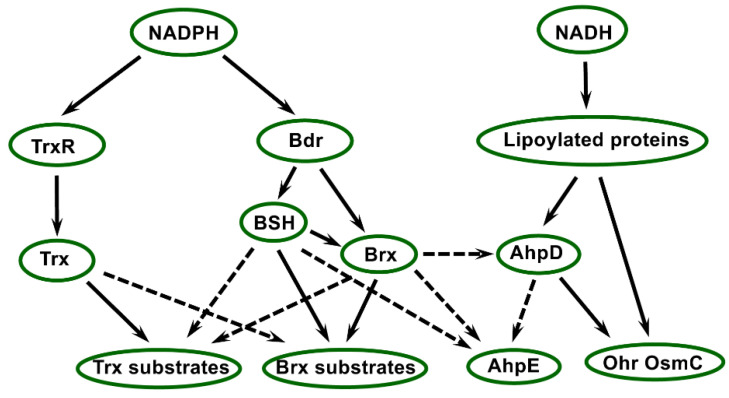
Relationships and possible interplays between *Deinococcus* thiol reductase systems. Dashed arrows indicate possible transfer of electrons between Trx-, BSH-, and lipoylated-dependent systems. TrxR, thioredoxin reductase; Trx, thioredoxin; Bdr, NADPH–BSH disulfide reductase; BSH, bacillithiol; Brx, bacilliredoxin; Ahp, alkyl hydroperoxide reductase; Ohr, organic hydroperoxide resistance enzyme; OsmC, osmotically induced bacterial protein C.

**Table 1 antioxidants-11-00561-t001:** Information of complete genomes of *Deinococcus* species.

Species (Abbreviation)	Identified in	Genome Size (Mb)	Replicons (Kb)	Proteins	References
*D. radiodurans* (DR)	Canned meat, USA	3.28	4 (2649, 412, 177, 46)	3167	[34,35,36]
*D. deserti* (Deide)	Sahara Desert sand, Morocco/Tunisia	3.86	4 (2820, 325, 314, 396)	3503	[5,37]
*D. geothermalis* (Dgeo)	Hot spring, Italy	3.25	3 (2467, 574, 206)	3003	[38,39]
*D. gobiensis* (DGo)	Gobi Desert sand, China	4.41	7 (3137, 433, 425, 232, 72, 55, 53)	4140	[40,41]
*D. maricopensis* (Deima)	Sonoran Desert soil, USA	3.5	1 (3499)	3242	[42,43]
*D. peraridilitoris* (Deipe)	Coastal desert soil, Chile	4.51	3 (3882, 557, 75)	4223	[44]
*D. proteolyticus* (Deipr)	Lama glama faeces, Japan	2.89	5 (2147, 315, 196, 132, 97)	2645	[35,45,46]

**Table 2 antioxidants-11-00561-t002:** Locus tags of genes encoding thiol reductases and related proteins from seven *Deinococcus* species.

Name/Description	DR	Deide	Dgeo	DGo	Deima	Deipe	Deipr
**Thioredoxin reductase, thioredoxins, and thioredoxin-like proteins**							
TrxR	_1982	_05800	_1576 _2772	_CA2339	_1454	_0175 _3902	_0873
TrxA (Trx1)	_0944	_18600	_1837	_CA0861	_2910	_3068 _3873 _3901	_0424
TrxC (Trx2)	_A0164	_01140	_2518				
Trx-like	_2085	_06390	_1508	_CA2073	_1013	_0695	_0576
Trx-like	_0057	_13741	_0729	_CA0407	_1186	_0565	_1451
Trx-like	_A0072		_2583	_PC0201 _PC0211			_2424 _2732
Trx-like	_B0110		_1776	_PA0204			_2190
Trx-like	_0948	_06780	_1960	_CA2541	_2994		_1792
**Other thiol-based disulfide oxidoreductases, (predicted) cytoplasmic**							
FrnE	_0659	_00690	_2073	_CA0380	_0892	_2202	_1901
FrnE-like		_3p01230					_1559
DSBA-like Trx domain-containing protein	_2335	_22890		_CA0030	_0620	_1019	
**Other thiol-based disulfide oxidoreductases, (predicted) periplasmic or cytoplasmic membrane**							
DsbA family protein	_2019 _0560	_06420	_0747	_CA1399	_1134		_0943
DsbA family protein	_0753	_12740	_0692	_CA1008	_1749		_0493 (and _2421?)
DsbB family	_0754	_12730	_0691	_CA1007	_1748		_0492
DsbD family; CcdA	_1300	_08350	_1241	_CA1639	_1155	_0794	_0982
TlpA-like family; DsbE/CcmG	_0345 _0189	_08290	_1248	_CA2017 _PC0193	_1148 (and _1627?)	_0801 _4366	_0892
DsbD family; CcdA		_2p00430				_0661	
TlpA-like family; DsbE/CcmG		_2p00420				_0660	
**Thioredoxin-dependent methionine sulfoxide reductases, cytoplasmic**							
MsrA	_1849	_10980	_0843	_CA1541	_1788	_3499	_1412
MsrB	_1378	_04050	_2072	_CA0919	_1441	_4299	_1900
**Potential Mo-dependent methionine sulfoxide reductases, predicted cytoplasmic**							
Molybdopterin oxidoreductase family protein	_0397	_18410	_0402	_CA0112	_0538	_1423	_1886
Sulfite oxidase family, molybdopterin-binding domain	_0716	_17540	_1719	_CA1115	_0813	_2833	_0695
**Mo-dependent methionine sulfoxide reductase system, periplasmic**							
MsrP	“_2536” (frame-shift)	_20380	_0877	_CA2733	_3114	_2978	_1129
MsrQ	_2537	_20370	_0878	_CA2734	_3115	_2977	_1128
**Thioredoxin-dependent peroxidases**							
PRX_BCP	_0846	_10900	DgeoAM_1323 ^a^	_CA1364	_2368	_0259	_0703
PRX_BCP	_1209 _1208	_09051	_0990 _2729	_CA1403	_1714 _0169	_3580 _3178	_1557
PRX_BCP		_23291 (partial?)		_CA0314			
**Alkyl hydroperoxide reductases**							
AhpE (PRX_AhpE_like)	_2242	_02430	_0122	_CA2657	_0618	_1016	_0175
AhpD-like	_1765	_13030 _1p00700	_1446	_CA1027	_0298	_3296 _4199 _3903 _3878 _3900	_2741
**OsmC/Ohr/YhfA family proteins**							
OsmC	_1538	_16090	_0526	_CA1241	_0667	_3743	
Ohr	_1857		_0446	_CA0901 _CA1828	_2331 _0137	_0225	_0815 _0816
YhfA	_1177	_10790 _21170	_1268	_CA1763	_2343	_0234 _0648	_0697
**Bacillithiol disulfide reductase, bacilliredoxin**							
Bdr	_2623	_23360	_2331	_CA0078	_0670	_2475	_1732
Brx (AbxC)	_1832	_14700	_1464	_CA1021	_1446	_3166	_0555

^a^ Not initially predicted; corresponds to protein WP_041221145.1.

**Table 3 antioxidants-11-00561-t003:** Active site motifs in *Deinococcus* Trx and Trx-like proteins.

	Trx1/Trx2	DR_2085-Type (TRP14-Like)	DR_A0072-Type	DR_B0110-Type	DR_0057-Type (MGP12-Like)	DR_0948-Type
**DR**	WCGPC	WCPDC	DCPDC	ACPGC	GCHLC	ECPGC
**Deide**	WCGPC	WCPDC			GCHLC	ECSGC
**Dgeo**	WCGPC	WCPDC	NCSSC	NCPAC	GCHLC	ECPGC
**DGo**	WCGPC	WCPDC	QCADC GCASC	ACPDC	GCHLC	ECPGC
**Deima**	WCGPC	WCPDC			GCHLC	ECAGC
**Deipe**	WCGPC	WCPDC			GCHLC	
**Deipr**	WCGPC	WCPDC	DCADC TCPDC	NCPNC	SCKLC	ECAGC

## Data Availability

The data presented in this study are available in review.

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
