# Peer review of "Thiol Reductases in Deinococcus Bacteria and Roles in Stress Tolerance"

_antioxidants, 2022, doi:10.3390/antiox11030561_

Round 1

Reviewer 1 Report

This review paper summarizes the action characteristics of thiol reductase represented by strains in Deinococcus, a radiation-resistant bacterium. Control of redox is an important stress response mechanism and is known as one of several strategies to cope with oxidative stress.
Electrons transferred from NADPH to TrxR are transferred to Trx, which is transferred to various substrates, causing various corresponding mechanisms.
The authors discovered and compared this process in the genomes of bacteria in the genus Deinococcus, and summarized the concept in general from a lot of reported information. Overall, it is worth noting that the complex pathways are clearly compared and organized around major proteins, such as TrxR, Trx, Dsb, Msr, BCP, Ahp, OsmC and Ohr.
However, the comparison with the case of eukaryotes (human, mice, plant) showing relatively low similarities is considered to be insufficient and over-expanded, so it is suggested to organize this part at the level of deletion or move into supplementary data.

It is desirable to correct the overall word in British English notation, and it is requested to correct it by following the opinion below.

Lines 356-366: How about getting rid of this paragraph or moving to supplementary materials including Fig. 2C.

Line 534: comma change to dot.

Please change Fig. 4C

Need to change in British English notation (maybe it is a guideline in the journal). For example, signaling > signaling (line 106/215), characterized > characterised (Z>S) in many sites.

Author Response

Reviewer 1's comments (in bold) followed by our replies.

- However, the comparison with the case of eukaryotes (human, mice, plant) showing relatively low similarities is considered to be insufficient and over-expanded, so it is suggested to organize this part at the level of deletion or move into supplementary data.

- Lines 356-366: How about getting rid of this paragraph or moving to supplementary materials including Fig. 2C.

In the manuscript, a comparison of Deinococcus thiol reductases with others from non-bacterial organisms is mainly made concerning Trx-like proteins, which form complex subfamilies present in specific phyla and for which the knowledge is still limited, notably in bacteria. We thus believe that this comparison brings valuable information about the possible physiological functions of Trx-like proteins in Deinococcus. This is particularly the case for TRP14- and MGP12-like proteins described in sections 2.3.1.1. and 2.3.2.1., which we think are not over-expanded. Therefore, we kept the initial versions of these sections.

- It is desirable to correct the overall word in British English notation, and it is requested to correct it by following the opinion below.

- Need to change in British English notation (maybe it is a guideline in the journal). For example, signaling > signaling (line 106/215), characterized > characterised (Z>S) in many sites.

We thoroughly checked the manuscript to use the British English notation; “signaling” has been corrected to “signalling”, “characterization” to “characterisation”, “stabilization” to “stabilisation”, ”oxidized” to “oxidised”, “harbor” to “harbour” “catalytical” to “catalytic”…

- Line 534: comma change to dot.

The comma has been deleted (line 539).

- Please change Fig. 4C

In the revised version, Fig. 4 has been moved to supplemental data. We did not modify the panel C in this figure since it is the original tree form that is generated when using UniProt ClustalO (https://www.uniprot.org/align/).

Reviewer 2 Report

Deinococcus bacteria have outstanding tolerance for the oxidative damage, and its protective mechanisms have been studied to understand how cells manage extreme oxidative stress. This review summarized recent studies about thiol reductases and proposed that the importance of the diversity of thiol peroxidases in Deinococcus for connecting various stress response pathways, such as UV radiation, metal stress, hyperosmotic stress etc. This review is informative and covers relevant topics. The following comments would be helpful by organizing figures and tables.

Comments:

The main message of this review is that thiol peroxidases in Deinococcus are linked to various stress response pathways. I recommend the authors to move the supplemental table S1 to the main. This table is referred many times in the manuscript, and it is helpful to see the diversity of that thiol peroxidases in Deinococcus. In exchange. Figures 2 and 4 can be moved to the supplementary.

Oxidative stress response via thiol peroxidases is connected to other stress responses in Deinococcus. This point is convincing, however why are cells remarkably tolerant when several stress response pathways are linked? If possible, I would like the authors to discuss this point.  

Trx-like proteins in Deinococcus are interesting candidates to investigate. Their active site motifs CXXC look unique and diverse. It would be nice if they are listed in a table for readers.

Line 159-161. “They concluded that four residues (M84, K137, F148 and F149) in D. radiodurans TrxR mainly account for the stability of the interface with Trx, and for the species-specific interaction.” What does ”species-specific interaction” mean?

Author Response

Reviewer 2's comments (in bold) followed by our replies.

- I recommend the authors to move the supplemental table S1 to the main. This table is referred many times in the manuscript, and it is helpful to see the diversity of that thiol peroxidases in Deinococcus. In exchange. Figures 2 and 4 can be moved to the supplementary.

As recommended, Table S1 in the first version has been integrated in the manuscript as Table 2 (pages 4 and 5) and Figure 4 has been moved to supplemental data (now Figure S11) (and we have renumbered figures accordingly). However, we kept Figure 2 in the manuscript since we think that Trx-like proteins are original thiol reductases, for which the knowledge is still poor, and that are well represented in Deinococcus (see answer to Reviewer 1’s related comment). Note that there is also a supplemental figure including five panels (Figure S5) related to Deinococcus Trx-like proteins.

- Oxidative stress response via thiol peroxidases is connected to other stress responses in Deinococcus. This point is convincing, however why are cells remarkably tolerant when several stress response pathways are linked? If possible, I would like the authors to discuss this point.

We modified the end of the last paragraph in Conclusions (lines 983-989) to highlight the possibility that the various pathways for electron transfer via thiol reductases and peroxidases in Deinococcus give rise to multiple connections and possible redundancy among them, and thus allow the establishment of protective and repair mechanisms that can substitute or be switched on differentially depending on stress conditions, leading to the implementation of a complex and robust network for maintaining protein redox homeostasis.

- Trx-like proteins in Deinococcus are interesting candidates to investigate. Their active site motifs CXXC look unique and diverse. It would be nice if they are listed in a table for readers.

A new table (Table 3, page 8) listing Deinococcus Trx active site motifs has been made and is referred line 264.

- Line 159-161. “They concluded that four residues (M84, K137, F148 and F149) in D. radiodurans TrxR mainly account for the stability of the interface with Trx, and for the species-specific interaction.” What does ”species-specific interaction” mean?

We modified the sentence (lines 168-169) to “They concluded that four residues (M84, K137, F148 and F149) in D. radiodurans TrxR mainly account for the stability of the interface and the preferential binding affinity towards D. radiodurans Trx2.”  This better explains its meaning, i.e. the involvement of the four residues in interface stability and preferential binding affinity of DR_TrxR towards DR_Trx2.